# The Potential Role of Neutrophil-Reactive Intensity (NEUT-RI) in the Diagnosis of Sepsis in Critically Ill Patients: A Retrospective Cohort Study

**DOI:** 10.3390/diagnostics13101781

**Published:** 2023-05-18

**Authors:** Elena Maria Alessandra Mantovani, Paolo Formenti, Stefano Pastori, Vincenzo Roccaforte, Miriam Gotti, Rossella Panella, Andrea Galimberti, Roberto Costagliola, Francesco Vetrone, Michele Umbrello, Angelo Pezzi, Giovanni Sabbatini

**Affiliations:** 1S.C. Anestesia, Rianimazione e Terapia Intensiva, ASST Nord Milano, Ospedale Bassini, 20097 Cinisello Balsamo, Italy; elena.mantovani@asst-nordmilano.it (E.M.A.M.); paolo.formenti@asst-nordmilano.it (P.F.); miriam.gotti@asst-nordmilano.it (M.G.); andrea.galimberti@asst-nordmilano.it (A.G.); roberto.costagliola@asst-nordmilano.it (R.C.); angelo.pezzi@asst-nordmilano.it (A.P.); 2S.C. Analisi Chimico Cliniche e Microbiologiche, ASST Nord Milano, Ospedale Bassini, 20097 Cinisello Balsamo, Italy; stefano.pastori@asst-nordmilano.it (S.P.); vincenzo.roccaforte@asst-nordmilano.it (V.R.); rossella.panella@asst-nordmilano.it (R.P.); 3S.C. Anestesia e Rianimazione II, ASST Santi Paolo e Carlo, Ospedale San Carlo, 20148 Milan, Italy; michele.umbrello@asst-santipaolocarlo.it

**Keywords:** sepsis, neutrophil reactivity (NEUT-RI), C-reactive protein, procalcitonin

## Abstract

The diagnosis of sepsis is often difficult and belated, substantially increasing mortality in affected patients. Its early identification allows for us to choose the most appropriate therapies in the shortest time, improving patients’ outcomes and eventually their survival. Since neutrophil activation is an indicator of an early innate immune response, the aim of the study was to evaluate the role of Neutrophil-Reactive Intensity (NEUT-RI), which is an indicator of their metabolic activity, in the diagnosis of sepsis. Data from 96 patients consecutively admitted to the Intensive Care Unit (ICU) were retrospectively analyzed (46 patients with and 50 without sepsis). Patients with sepsis were further divided between sepsis and septic shock according to the severity of the illness. Patients were subsequently classified according to renal function. For the diagnosis of sepsis, NEUT-RI showed an AUC of >0.80 and a better negative predictive value than Procalcitonin (PCT) and C-reactive protein (CRP) (87.4% vs. 83.9% and 86.6%, *p* = 0.038). Unlike PCT and CRP, NEUT-RI did not show a significant difference within the “septic” group between patients with normal renal function and those with renal failure (*p* = 0.739). Similar results were observed among the “non-septic” group (*p* = 0.182). The increase in NEUT-RI values could be useful in the early ruling-out of sepsis, and it does not appear to be influenced by renal failure. However, NEUT-RI has not proved to be efficient in discriminating the severity of sepsis at the time of admission. Larger, prospective studies are needed to confirm these results.

## 1. Introduction

The progression of sepsis into septic shock may be reduced by its early detection, reducing the risk of death [1]. At the same time, its accuracy is desirable to avoid the indiscriminate use of antimicrobial therapy that leads to antimicrobial resistance [2,3]. Currently, markers of sepsis such as white blood cell count (WBC) and C-Reactive Protein (CRP) are commonly used in critical care illness despite their non-specificity for sepsis [4]. The evaluation of biological markers such as procalcitonin (PCT) [5], tumor necrosis factor [6] and Interleukin-6 [7] may be helpful, but their costs can be significantly high, and the results could be delayed due to the processing time. Microbiological cultures are currently the gold standard for the detection of many pathogens, allowing for a diagnosis of sepsis [8]. However, their processing time takes at least 24 h, they have low sensitivity and frequently are at risk of contamination [9]. A Sequential Organ Failure Assessment (SOFA) score of ≥2 (required by the current definition of sepsis) represents a global mortality risk, and >10% represents hospital admission [10]. Thus, it is difficult to decide if critically ill patients are septic. Recently, understanding the role of neutrophil granulocytes in inflammation has broadly changed [11]. The understanding that activated neutrophils can react to most of the macrophage’s functions replaced the previous vision that they play a passive role when reacting to external signals. In fact, once activated, the neutrophils produce a variety of pro-inflammatory cytokines and coat molecules, thus permitting the presentation of antigen to lymphocytes with their subsequent activation [12]. A routinary analysis of WBC by a flow-cytometry-based method was shown to be quick, costless and was included as part of the routine full blood count analysis [13]. Once an infection occurs, less mature neutrophil forms enter the circle, leading to a shift between a higher immature/total granulocyte ratio and a higher neutrophil band count [14]. Clinical studies on septic ICU patients already showed a correlation between the activation of neutrophils and monocytes [15]. The use of a fluorescence-flow cytometer was shown to be feasible to detect a real-time study of monocyte and neutrophil activation [16]. Moreover, this method was able to show information about cell shape, cell membrane content and cell granularity, with an overall description of the WBC activation status [17]. In fact, since these cells actively produce pro-inflammatory signals, they have greater activity in the cytoplasm, leading to greater intensity in the fluorescence signal of activated cells compared to the resting one. Neutrophil Granularity Intensity (NEUT-GI) and the Neutrophil-Reactive Index (NEUT-RI) represent indicators of an early innate immune response [18,19,20]. NEUT-GI reflects the increase in inflammatory processes since it assesses the cytoplasmic neutrophil’s granularity. NEUT-RI reflects the metabolic activity of a neutrophil population by measuring the fluorescence intensity (FI) [19]. Since neutrophil activation is an indicator of an early innate immune response, NEUT-RI may correlate with sepsis [21]. It could predict the appearance of inflammatory markers such as immature granulocytes, thus detecting bacterial infection earlier. More specifically, an increase of >51.00 FI NEUT-RI correlated with an increase of 2.4% of immunoglobulin over 72 h after the infection [18,22]. Similarly, it was significantly greater in patients with post-burn injury sepsis as compared with patients without sepsis, suggesting potential for the early diagnosis of sepsis [19]. The aim of this study was to retrospectively evaluate sepsis’ diagnostic performance of NEUT-RI as compared with PCT and CRP.

## 2. Materials and Methods

This was a retrospective observational study conducted on patients consecutively admitted to the Intensive Care Unit of two Italian city hospitals between March and November 2022. The inclusion criterion was being admitted to the ICU with any diagnosis. The exclusion criteria were: age <18, diagnosis of current malignancy, chronic corticosteroid therapy (prednisone >10 mg/die or equivalent), immunosuppressive therapy and congenital immunodeficiency. The study protocol was approved by the local Ethical Review Board (n. S00081/2022). Written informed consent was waived because of the retrospective nature of the study and data were treated anonymously according to the General Data Protection Regulation-GDPR UE 679/2016. Patients were retrospectively divided in “septic” and “not septic” groups by the authors as defined by Sepsis-III criteria (patients with sepsis had a suspected infection [23] and evidence of organ dysfunction with SOFA >2 [24]). Patients with sepsis were further divided between patients with sepsis and patients with septic shock according to the septic shock diagnosis (Lactate >2 mmol/L and vasopressors were required to maintain the mean arterial pressure >65 mmHg) [1]. The diagnosis was retrieved from ICD-9-CM codes at admission on the electronic clinical registry. Demographics, WBC count, CRP, PCT, NEUT-RI, serum creatinine, blood cultures results and SOFA score at the time of ICU admission were collected for patients in both “septic” and “not septic” groups. To evaluate the effect of renal function on the sepsis biomarkers, patients were then divided into “renal failure” (including acute kidney injury—AKI, and chronic renal disease—CRD) and ”normal renal function” groups according to the KDIGO (Kidney Disease Improving Global Outcomes) classification [25,26]. CRP, PCT and Creatinine were measured on serum using the automated clinical chemistry analyzer Beckman Coulter AU 5800 and Beckman Coulter UniCel DxI800 immunochemical analyzer (Beckman Coulter, Brea, CA, USA), according to manufacturer’s recommendations. The PCT was measured by chemiluminescent CRP using the turbidimetric method and creatinine by the enzyme immunoassay.

The complete blood cell count was performed using a Sysmex XN hematology analyzer (Sysmex, Kobe, Japan), which enumerates and classifies blood cells by the means of flow cytometry. The XN Series hematology analyzers aid in the identification of NEUT-RI, whose signal differs significantly from that of quiescent cells. Fluorescence flow cytometry analysis allows for the measurement of cellular function as part of a routine hematological examination. The positioning of the neutrophils within the scattergram allows for us to evaluate the activation of the neutrophils. In particular, the NEUT-RI parameter reflects the intensity of neutrophil reactivity, expressed as fluorescence intensity (FI).

Data were collected anonymously into a Microsoft Excel spreadsheet. Statistical analyses were performed with GraphPad v. 6.0 and SPSS v. 2.8 statistical packages. ROC curve analysis was performed with STATA 14.0 (Stata Corp., College Station, TX, USA). The results were expressed as mean ± standard deviation (SD) or median (interquartile range—IQR) where appropriate. Mann–Whitney or Kolmogorov–Smirnov tests, if variables were not normally distributed, were applied for estimating the differences between groups and subgroups. Diagnostic performance of the biomarkers was evaluated with ROC analysis. The best cutoff values were selected according to the Youden test. The results were applied for the calculation of positive and negative predictive values of the tests, which were compared using the McNemar’s test. The areas under the ROC curves were compared with the STATA command “roccomp”. A *p*-value ≤ 0.05 was considered statistically significant. We calculated that a sample size of at least 92 patients would allow for us to observe an area under the ROC curve of 0.85 with a power of 0.8 and an alpha of 0.05 [27].

## 3. Results

### 3.1. Emographic Caracteristics and Subgroup Analysis

During the inclusion period, a total of 118 patients were screened for inclusion in the study. A total of 22 patients were excluded according to the exclusion criteria, leaving a total of 96 patients enrolled in the study. The main characteristics of the patients are summarized in Table 1. A total of 46 patients (22 females and 24 males) were included in the “septic” group (47.9%), while 50 (22 females and 28 males) were in the “non-septic” group (52.1%). The patients who were “septic” were categorized in the sepsis (*n* = 21, 45.7%) or septic shock (*n* = 25, 54.3%) subgroups according to sepsis severity at the time of admission to the ICU. Among the “septic” group, blood cultures samples were positive in 15 and negative in 14 patients (they were not collected in 17 patients). More than half of the patients (51%) showed renal failure at the time of admission to the ICU. In the “septic” group, 28 patients had renal failure at admission (60.9%); the mean value of normal plasmatic creatinine was 0.89 mg/dL (±SD 0.21), while in those presenting renal failure at admission, it was 3.51 mg/dL (±SD 2.35) (*p* < 0.001). In the “non-septic” group, 21 patients had renal failure (42%); the mean value of normal renal function creatinine was 0.83 mg/dL (±SD 0.26), while in those presenting renal failure, it was 2.04 mg/dL (±SD 2.25) (*p* < 0.001). Figure 1 describes the study design.

### 3.2. Performance Evaluation of Inflammatory Parameters

The values of NEUT-RI, PCT and CRP were evaluated in “septic” and “non-septic” patients. NEUT-RI were analyzed in all patients, showing a significant difference between the two groups (57 [52.8; 62.7] FI vs. 48.7 [47.1; 51.7] FI, respectively, *p* < 0.001) and AUC values (0.826). Considering the value of NEUT-RI ≥ 51.9 FI was the best cut-off value, the sensitivity of NEUT-RI was 80.4%, while the specificity was 76%. From the regional report of infections of patients admitted to ICUs in 2021, the average prevalence of sepsis was 35.89%. Consequently, in the study population, the positive predictive value of NEUT-RI for the detection of sepsis is 65.2%, while the negative predictive value is 87.4%. Regarding the evaluation of the performance of NEUT-RI for the discrimination of sepsis severity, no significant difference was found between the sepsis and septic shock subgroups (56.97 [47.1; 88.6] FI vs. 63 [41; 112] FI, respectively, *p* = 0.075). Regarding the inflammatory parameter PCT, 81 values were collected. There was a significant difference between “non-septic” and “septic” groups (0.48 [0.9; 1.27] ng/mL vs. 17.7 [7.8; 74.5] ng/mL, *p* < 0.001) with AUC values of 0.855. Considering a PCT value ≥ 2.16 ng/mL was the best cut-off value, in our study population, PCT sensitivity was 69.6%, while specificity was 88.6%, with a positive predictive value of 77.3% and a negative predictive value of 83.9%. Concerning the evaluation of PCT performance for the discrimination of sepsis severity, a significant difference was observed between sepsis and septic shock subgroups (3.22 [1.63; 17.22] ng/mL vs. 64.2 [12.4; 113] ng/mL, *p* < 0.001). As for the evaluation of CRP performance for the detection of sepsis, 87 values were analyzed, showing a significant difference between “non-septic” and “septic” group (3.3 [1.43; 11.2] mg/dL vs. 18.1 [8.3; 25.3] mg/dL, *p* < 0.001). Considering a CRP ≥ 6.91 mg/dL was the best cut-off value, in the study population, CRP sensitivity was 80.4%, while specificity was 70.7%, with a positive predictive value of 60.6% and a negative predictive value of 86.6%. Regarding the evaluation of CRP performance for the discrimination of sepsis severity, a significant difference was observed between sepsis and septic shock subgroups (12.7 [5.3; 18.3] mg/dL vs. 18.7 [17.3; 27.9] mg/dL, *p* = 0.018). Table 2 summarized the accuracy (AUC), cut-off, sensitivity and specificity of inflammatory parameters for the detection of sepsis. The comparisons between the negative predictive values of each parameter showed higher values for NEUT-RI (*p* = 0.038), while the comparisons between the positive predictive values of each parameter showed higher values for PCT (*p* = 0.021) (Figure 2).

### 3.3. Renal Failure Influence on Inflammatory Markers

We compared the values of each biomarker in relation to renal function within each group (“septic” versus “non-septic”), as shown in Table 3. NEUT-RI did not show a significant difference within the “septic” group between patients with normal renal function and those with renal failure (AKI and/or CKD) at admission to the ICU (*p* = 0.739). Among the “non-septic” group, no significant difference was observed between the two subgroups as well (*p* = 0.182). PCT showed a significant difference between patients with normal renal function and those with renal failure (AKI and/or CKD) at admission to the ICU (*p* = 0.002 and 0.016, respectively) within “septic” and “non-septic” groups. Similarly, CRP showed a significant difference between patients with normal renal function and those with renal failure (AKI and/or CKD) at admission to the ICU (*p* = 0.005) within the “septic” group. Differently, in the “non-septic” group, a significant difference between patients with normal renal function and those with renal failure (AKI and/or CKD) (*p* = 0.162) was not observed.

## 4. Discussion

The main findings of the present investigation can be summarized as follows: (1) NEUT-RI was higher in critically ill patients with sepsis than in those who were hospitalized for other causes; (2) PCT was the biomarker with the best positive predictive value as compared to NEUT-RI and CRP; (3) however, NEUT-RI had the best negative predictive value compared to PCT and CRP, suggesting its diagnostic use to minimize false negatives and increasing the detection of patients with sepsis; (4) NEUT-RI was not significantly different between patients with and without renal failure.

Currently, it is of primary importance to be able to differentiate patients with sepsis in hospital wards [3]. First, clinicians should distinguish between inflammations or non-inflammations caused by infections. Then, in case of infection, the status of the immune response and the pathogen responsible should be investigated. Early information about inflammatory reactions may be derived by hematological inflammation parameters. Routine laboratory tests allow for a combination of the complete blood count and novel diagnostic inflammation biomarkers [18]. These parameters may quantify and distinguish the activation status of white blood cell sub-populations, providing additional information about the activation of the immune response [28]. Moreover, humoral immune response, adaptive cell-mediated response and early innate response can be differentiated by the diagnostic inflammation biomarkers [20]. Differentiating between infections and inflammation, and between different pathogenic causes of infection, allows for the management of inflammatory diseases. Our results showed that inflammatory markers, including WBC, CRP, PCT and NEUT-RI, were independently associated with sepsis in critically ill patients admitted to the ICU. The early detection of sepsis is necessary so that specific goal-directed therapy bundles to reduce complications may begin [29]. Thus, many studies have focused on early sepsis detection, using biomarkers and scoring systems [30,31]. Recently, an increasing number of studies have shown how NEUT-RI was significantly higher in patients with sepsis as compared with patients who were non-septic [32,33]. Even if microbial cultures are still necessary to define sepsis detection, their results are not always definitive and are usually obtainable after several days [34]. In addition, in patients receiving antimicrobial treatment, the results sometimes provide a false negative. Our results showed how NEUT-RI has a negative predictive value superior to CRP and PCT. It is recognized that CRP is an acute inflammatory marker in the acute-phase reaction of sepsis [4,35,36]. Our results confirmed the previous results, showing a sensibility of 80.4% and specificity of 70.7%, with the best cutoff value of 6.91 mg/dL. Similarly, PCT was shown to have high accuracy in sepsis diagnosis; it started to rise a few hours after systemic inflammation, making it valuable in the early detection of sepsis [5,37,38]. In our study, PCT confirmed high specificity in recognition of patients with sepsis, showing a greater positive prognostic value as compared with CRP and NEUT-RI. Our results showed how CRP and PCT were affected by impaired renal function. CRP is known to be negatively correlated with the glomerular filtration rate [39]; nevertheless, it can properly predict infection in patients with impaired renal function [40,41]. PCT is eliminated by kidney clearance; thus, high concentrations of PCT in patients with acute kidney dysfunction may result in elevation, even in the absence of infection [42,43]. Thus, its sensitivity for the diagnosis of bacterial infection may be lower, even if the best cut-off value in patients with acute kidney injury remains unknown and the real relation among creatinine/urea and PCT concentration are not clear [37]. In this case, NEUT-RI appears to be more effective. Its reading was not influenced by primary diseases such as liver and kidney diseases.

This study has several limitations. First, it was not always possible to compare NEUT-RI with PCT and CRP in the first hours after the onset of sepsis (in the emergency room or in the hospital ward), since the latter were not always requested, and were only analyzed at admission to the ICU. Furthermore, in five patients with sepsis, PCT was higher than 250 ng/mL; therefore, it was necessary to approximate it to 250 ng/mL to allow for the statistical analyses. Patients without sepsis admitted to the ICU, PCT and CRP were not always analyzed, unlike NEUT-RI, which was already available in the blood count. For this reason, the “non-septic” values used to evaluate the diagnostic performance of PCT and CRP are fewer than those of NEUT-RI. We decided to consider the renal function as a parameter for sub-analysis since it strongly influences the sepsis biomarkers as mentioned above. Even if NEUT-RI was not significantly different between patients with and without renal failure, it cannot be translated into “no effect”. As a secondary objective, the results might be due to the limited power of the study, or it may be that the effect is small and not detectable with the sample size used. To the best of our knowledge, NEUT-RI analysis is currently only available on Sysmex XN hematology analyzer. Thus, our results, even if reproducible, might not be largely available to the clinicians. Eventually, our results could accelerate its integration in other systems to ensure the generalizability of our findings. Finally, our study did not consider the possible correlation of NEUT-RI and the infection biomarkers with 28-day mortality after admission to the ICU, leaving an open field for further investigations.

## 5. Conclusions

The diagnosis of sepsis is often difficult and late, substantially increasing mortality. Its early identification allows for us to implement the most appropriate therapies in the shortest time, improving patient outcomes and survival. The inflammatory biomarkers analyzed in this study were proven to be effective in supporting clinicians in the early diagnosis of sepsis. NEUT-RI provides a quantitative assessment of the activation status of neutrophils and early innate immune response, allowing for us to distinguish between an infectious and non-infectious inflammatory state. The increase in the NEUT-RI values could be useful in the early diagnosis of sepsis, in association with the clinical signs and inflammatory biomarkers currently used, and it would not appear to be influenced by renal failure. Moreover, since the complete blood count is the most broadly performed rapid laboratory investigation, and since NEUT-RI is already integrated into the blood count, it provides complementary information, allowing for us to rule out infection. However, NEUT-RI has not been proven to be efficient in discriminating the severity of sepsis at admission. Larger studies are needed to confirm these results.

## Figures and Tables

**Figure 1 diagnostics-13-01781-f001:**
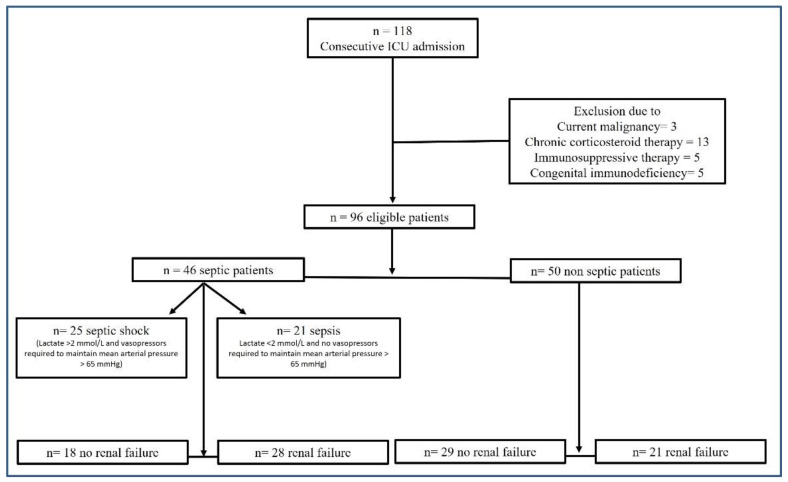
Study design flow chart.

**Figure 2 diagnostics-13-01781-f002:**
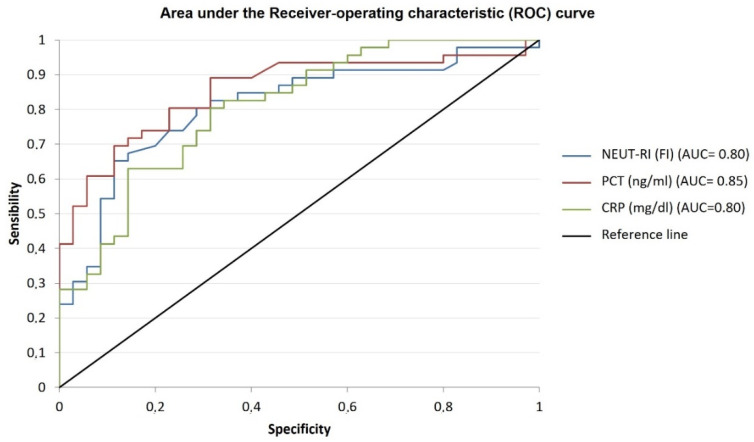
Performance evaluation of inflammatory parameters. The figure shows the area under the receiver operating characteristic (ROC) curve for the distinction of inflammatory parameters for the detection of sepsis. The areas under the ROC curves are as follows: NEUT-RI (blue line): 0.80 [95% CI 0.74–0.91]; PCT (red line): 0.85 [95% CI 0.77–0.93]; CRP (green line): 0.80 [95% CI 0.77–0.93; *p* < 0.001. NEUT-RI = Neutrophil-Reactive Index; PCT = procalcitonin; CR = C-reactive protein.

**Table 1 diagnostics-13-01781-t001:** Baseline characteristics of the study population, divided by the diagnosis of sepsis at ICU admission.

	Septic (*n* 46)	Non-Septic (*n* 50)	*p*
Age (years)	70 [46; 87]	68 [29; 90]	0.172
Male sex, *n* (%)	24 (56)	28 (58)	0.992
Septic shock *n* (%)	25 (54.3%)		
Diagnosis:			
Pneumonia, *n* (%)	23 (50)		
Peritonitis, *n* (%)	12 (26)		
Urinary tract infection, *n* (%)	11 (23)		
Coma		11 (22)	
Other neurologic disorders		14 (28)	
Acute pulmonary edema		7 (14)	
Post-surgery monitoring		18 (36)	
SOFA score at ICU admission (points)	7 [4; 8]	6 [4; 8]	0.951
Renal failure at ICU admission, *n* (%)	28 (60.1)	21 (42)	0.245
Serum creatinine (mg/dL)	3.51 (2.35)	2.04 (2.25)	<0.001
NEUT-RI (FI)	57 [52.8; 62.7]	48.7 [47.1; 51.7]	<0.001
PCT (ng/mL)	17.7 [7.8; 74.5]	0.48 [0.9; 1.27]	<0.001
CRP (mg/dL)	18.1 [8.3; 25.3]	3.3 [1.43; 11.2]	<0.001

**Table 2 diagnostics-13-01781-t002:** Accuracy, cut-off, sensitivity and specificity of inflammatory parameters for the detection of sepsis.

	AUROC (95% CI)	Cut-Off	Youden’s Index	Sens (95% CI)	Spec (95% CI)	PPV	NPV
NEUT-RI	0.80 [0.741–0.912]	≥51.9 FI	0.56	80.4% [68.9–91.8]	76% [64.2–87.8]	65.2%	87.4%
PCT	0.855 [0.771–0.938]	≥2.16 ng/mL	0.58	69.6% [56.3–82.9]	88.6% [78–99.1]	77.3%	83.9%
CRP	0.801 [0.736–0.908]	≥6.91 mg/dL	0.51	80.4% [68.9–91.9]	70.7% [56.8–84.7]	60.6%	86.6%

AUROC = area under the ROC curve; Sens = sensibility; Spec = specificity; PPV = positive predictive value; NPV = negative predictive value; NEUT-RI = Neutrophil-Reactive Index; PCT = procalcitonin; CRP = C-reactive protein.

**Table 3 diagnostics-13-01781-t003:** Inflammatory parameters in septic vs. non-septic patients, and in patients diagnosed with or without renal failure.

	Normal Renal Function	*n*	Renal Failure	*n*	*p*
NEUT-RI (septic)	52.3 [49.5; 58.3] FI	18	57.5 [55.1; 63.9] FI	28	0.182
NEUT-RI (non-septic)	49.3 [47; 52.7] FI	29	48.4 [47.6; 50.7] FI	21	0.739
PCT (septic)	1.7 [0.79; 5.3] ng/mL	18	57.9 [14.3; 107.3] ng/mL	28	0.002
PCT (non-septic)	0.43 [0.3; 0.7] ng/mL	20	1.23 [0.5; 6.8] ng/mL	15	0.016
CRP (septic)	9.13 [4.9; 17.2] mg/dL	18	18.7 [15.7; 27.7] mg/dL	28	0.005
CRP (non-septic)	2.7 [1.2; 4.7] mg/dL	24	6.9 [2.5; 11.6 ] mg/dL	17	0.162

NEUT-RI = Neutrophil-Reactive Index; PCT = procalcitonin; CRP = C-reactive protein.

## Data Availability

The data presented in this study are available on request from the corresponding author.

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
