# Peer review of "The Potential Role of Neutrophil-Reactive Intensity (NEUT-RI) in the Diagnosis of Sepsis in Critically Ill Patients: A Retrospective Cohort Study"

_diagnostics, 2023, doi:10.3390/diagnostics13101781_

Round 1
Reviewer 1 Report
The overall work described in the manuscript is good. But some certain changes and queries need to be addressed:
1. The full forms of the abbreviations used should be stated at the very first use of it in the manuscript as a rule of thumb. So, the authors should rectify that. For example, full forms of PCT and CRP in line no.24, full forms of FI and SOFA in line no. 76 and 93, respectively.
2. Why is only renal dysfunction considered a parameter in the study? While there are a range of co-morbidities associated with sepsis and inflammation.
3. The study design consisting of several groups and sub-groups of patients admitted in ICUs should be represented in an illustrative manner with flow charts, etc., for ease of understanding by the readers.
4. I am not satisfied with the concluding remarks and take-home message of the work. How this work will contribute to the already vast knowledge base available related to the diagnosis of sepsis? And what is the fate of NEUT-RI in this regard?
5. Since neonatal sepsis is widely prevalent, is there any data related to diagnostic markers of such concerning NEUT-RI? The author's take on this regard should be included in the conclusion.
At some points, the reading flow of the paper seems vague and out of order. Correcting the flow of paragraphs by using appropriate conjunctions and prepositions is suggested.
Author Response
We thank the reviewer for his/her comments and suggestions. Below you can find a point by point answers.
- The full forms of the abbreviations used should be stated at the very first use of it in the manuscript as a rule of thumb. So, the authors should rectify that. For example, full forms of PCT and CRP in line no.24, full forms of FI and SOFA in line no. 76 and 93, respectively.
A: We thank the reviewer for this observation. We carefully checked the paper abbreviations, ensuring to clarify all of them.
- Why is only renal dysfunction considered a parameter in the study? While there are a range of co-morbidities associated with sepsis and inflammation.
A: we thank the reviewer for this insightful comment. We considered renal dysfunction since other biomarkers of sepsis are influenced by this organ failure. We added this point in the limitations’ part, that now reads as:
“This study has several limitations. First, it was not always possible to compare NEUT-RI with PCT and CRP in the first hours after the onset of sepsis (in the emergency room or in the hospital ward) since the latter were not always requested but they were analyzed only at admission to the ICU. Furthermore, in 5 septic patients PCT was higher than 250 ng/mL, therefore it was necessary to approximate it to 250 ng/mL to allow the statistical analyses. Considering non-septic patients admitted to the ICU, PCT and CRP were not always analyzed, unlike NEUT-RI which was already available in the blood count in-stead. For this reason, the “non-septic” values used to evaluate the diagnostic perfor-mance of PCT and CRP are fewer than those of NEUT-RI. We decided to consider the renal function as a parameter for sub-analysis since it strongly influences the sepsis biomarkers as mentioned above. To the best of our knowledge NEUT-RI analysis is currently only available on Sysmex XN hematology analyzer Thus, our results even if reproducible might not be largely available to the clinicians. Eventually, our results could accelerate its integration in other systems to ensure the generalizability of our findings. Finally, our study did not consider the possible correlation of NEUT-RI and the infection biomarkers with 28-days-mortality after admission to the ICU, leaving an open field for further investigations”.
- The study design consisting of several groups and sub-groups of patients admitted in ICUs should be represented in an illustrative manner with flow charts, etc., for ease of understanding by the readers.
A: we thank the reviewer again, and we added a new figure (figure 1) that now describe the flow chart of the enrollment.
- I am not satisfied with the concluding remarks and take-home message of the work. How this work will contribute to the already vast knowledge base available related to the diagnosis of sepsis? And what is the fate of NEUT-RI in this regard?
A: we thank again the reviewer for his/her suggestion. We modified the conclusions and tried to define better its potential role in the sepsis diagnosis. Now the conclusions read as:
“The diagnosis of sepsis is often difficult and late, substantially increasing the mortality. Its early identification allows to implement the most appropriate therapies in the shortest time, improving the outcome and the survival. The inflammatory biomarkers analyzed in this study proved to be effective in supporting clinicians in the early diag-nosis of sepsis. In particular, NEUT-RI provides a quantitative assessment of the acti-vation status of neutrophils and early innate immune response, allowing to distinguish between an infectious and non-infectious inflammatory state. The increase in the NEUT-RI values could be useful in the early diagnosis of sepsis, in association with the clinical signs and inflammatory biomarkers currently used, and it would not appear to be influenced by renal failure. Moreover, since the complete blood counts is the most broadly performed rapid laboratory investigation, and since the NEUT-RI is already integrated into the blood count, it provides complementary information allowing to rule out infection. However, the NEUT-RI has not proved efficient in discriminating the severity of sepsis at admission. Larger studies are needed to confirm these results.”.
- Since neonatal sepsis is widely prevalent, is there any data related to diagnostic markers of such concerning NEUT-RI? The author's take on this regard should be included in the conclusion.
A: we thank the review for this observation. To our knowledge, there is few data regarding the use of NEUT-RI in neonatal sepsis diagnosis. Furthermore, the population of our study doesn’t include neonatal patients, so we do not believe a comment on it would be appropriate.
Reviewer 2 Report
General comments
=============
I appreciated the opportunity to peer-review your work on The potential role of Neutrophil-Reactive Intensity (NEUT-RI) in the diagnosis of sepsis in critically ill patients: a retrospective cohort study. This manuscript was well written.
Specific comments
=============
Major comments
---------------------
1. In the abstract (lines 24-25), the authors mention a better negative predictive value for NEUT-RI compared to PCT and CRP (87.4% vs. 83.9% and 86.6%, p<0.001). However, these findings are not discussed within the manuscript. Please provide the corresponding results and discussion sections to support this claim.
2. The abstract also states that NEUT-RI is not positively influenced by renal failure (p=0.212) (lines 25-26), but this content is not found in the manuscript. Furthermore, a statistically insignificant result does not necessarily imply a lack of positive influence. Please include this information in the manuscript and clarify the interpretation of these findings.
3. The introduction (lines 66-74) lacks scientific literature on NEUT-RI. Please provide relevant references to support the information presented in this section.
4. The manuscript focuses on NEUT-RI using the Sysmex XN hematology analyzer. Please provide an explanation of how NEUT-RI can be measured using other systems to ensure the generalizability of your findings.
5. The authors set a ROC curve of 0.85 (line 125) as the threshold for diagnostic performance. Please provide scientific literature to justify the choice of this threshold.
6. In the methods section (line 130), the authors mention that 22 patients were excluded from the study, but the reasons for their exclusion are not provided. Please specify the exclusion criteria for these patients.
7. The discussion (line 224) states that PCT is the biomarker with the best positive predictive value compared to NEUT-RI. To support this conclusion, the authors should provide a direct comparison between PCT, CRP, and NEUT-RI, demonstrating that PCT has a significantly higher AUC.
8. Before the conclusion, please include a section discussing the limitations of this study to provide a more balanced perspective on the findings and their implications.
9. The conclusion (lines 288-289) should be elaborated to better reflect the results and discussion sections. If the authors wish to conclude that NEUT-RI values could be useful in the early diagnosis of sepsis and are not influenced by renal failure (not only statistically insignificance), they should provide more supporting evidence and discussion on this topic.
Minor editing of English language required
Author Response
We thank the reviewer for his/her comments and suggestions.
- In the abstract (lines 24-25), the authors mention a better negative predictive value for NEUT-RI compared to PCT and CRP (87.4% vs. 83.9% and 86.6%, p<0.001). However, these findings are not discussed within the manuscript. Please provide the corresponding results and discussion sections to support this claim.
A: we thank the reviewer for this observation. We added this part into the results and discussion as suggested.
- The abstract also states that NEUT-RI is not positively influenced by renal failure (p=0.212) (lines 25-26), but this content is not found in the manuscript. Furthermore, a statistically insignificant result does not necessarily imply a lack of positive influence. Please include this information in the manuscript and clarify the interpretation of these findings.
A: we thank again the reviewer for this observation. In fact, the meaning of the sentence was wrong. We would indicate that NEUT-RI is not influenced by renal impairment, which is one the major strength of the parameters. We modified the text according to this.
- The introduction (lines 66-74) lacks scientific literature on NEUT-RI. Please provide relevant references to support the information presented in this section.
A: We added relevant the few references available to support the information presented in this section as suggested by the reviewer.
- The manuscript focuses on NEUT-RI using the Sysmex XN hematology analyzer. Please provide an explanation of how NEUT-RI can be measured using other systems to ensure the generalizability of your findings.
A: we thank the reviewer for this observation. To our knowledge, Sysmex XN hematology analyzer is the only available analyzer for NEUT-RI analysis. We added this comment in the limitations section, which now reads as:
“This study has several limitations. First, it was not always possible to compare NEUT-RI with PCT and CRP in the first hours after the onset of sepsis (in the emergency room or in the hospital ward) since the latter were not always requested but they were analyzed only at admission to the ICU. Furthermore, in 5 septic patients PCT was higher than 250 ng/mL, therefore it was necessary to approximate it to 250 ng/mL to allow the statistical analyses. Considering non-septic patients admitted to the ICU, PCT and CRP were not always analyzed, unlike NEUT-RI which was already available in the blood count in-stead. For this reason, the “non-septic” values used to evaluate the diagnostic perfor-mance of PCT and CRP are fewer than those of NEUT-RI. We decided to consider the renal function as a parameter for sub-analysis since it strongly influences the sepsis biomarkers as mentioned above. To the best of our knowledge NEUT-RI analysis is currently only available on Sysmex XN hematology analyzer Thus, our results even if reproducible might not be largely available to the clinicians. Eventually, our results could accelerate its integration in other systems to ensure the generalizability of our findings. Finally, our study did not consider the possible correlation of NEUT-RI and the infection biomarkers with 28-days-mortality after admission to the ICU, leaving an open field for further investigations”.
- The authors set a ROC curve of 0.85 (line 125) as the threshold for diagnostic performance. Please provide scientific literature to justify the choice of this threshold.
A: we added relevant scientific literature to justify the choice of this threshold.
- In the methods section (line 130), the authors mention that 22 patients were excluded from the study, but the reasons for their exclusion are not provided. Please specify the exclusion criteria for these patients.
A: as suggested by the reviewer #1, we specified it creating a new flow chart figure of the study design, in which the exclusion criteria for those patients were described.
- The discussion (line 224) states that PCT is the biomarker with the best positive predictive value compared to NEUT-RI. To support this conclusion, the authors should provide a direct comparison between PCT, CRP, and NEUT-RI, demonstrating that PCT has a significantly higher AUC.
- we thank the reviewer for this observation. We completed the analysis and added it into the results and discussion sections. In particular, the predictive values were directly compared using the McNemar’s test, whereas the areas under the ROC curves were compared with the STATA command “roccomp”, wich uses a non-parametric approach using an algorithm suggested by DeLong et al. (DeLong, Biometrics 1988).
- Before the conclusion, please include a section discussing the limitations of this study to provide a more balanced perspective on the findings and their implications.
A: we thank the reviewer for this observation. A limitation part was already present. We modified it according also to the reviewer’s #1 observation.
- The conclusion (lines 288-289) should be elaborated to better reflect the results and discussion sections. If the authors wish to conclude that NEUT-RI values could be useful in the early diagnosis of sepsis and are not influenced by renal failure (not only statistically insignificance), they should provide more supporting evidence and discussion on this topic.
A: we modified this section according to this observation and the others suggested by reviewer #1
Round 2
Reviewer 2 Report
General comments
=============
I appreciated the opportunity to peer-review your work on The potential role of Neutrophil-Reactive Intensity (NEUT-RI) in the diagnosis of sepsis in critically ill patients: a retrospective cohort study. This manuscript was well written. Almost all responses were reasonable.
Specific comments
=============
Major comments
---------------------
1. My primary concern lies within the assertion made in lines 255-256 where you state, "4) the NEUT-RI was not affected by renal failure." The authors need to delineate the basis of this claim more explicitly. In line with my previous comment No.2, it's essential to understand that a statistically insignificant result does not necessarily translate to "no effect" or "not affected". The results might be due to the limited power of the study, or it may be that the effect is small and not detectable with the sample size used. Therefore, it's crucial to provide the reasoning.
Author Response
General comments
=============
I appreciated the opportunity to peer-review your work on The potential role of Neutrophil-Reactive Intensity (NEUT-RI) in the diagnosis of sepsis in critically ill patients: a retrospective cohort study. This manuscript was well written. Almost all responses were reasonable.
Specific comments
=============
Major comments
---------------------
- My primary concern lies within the assertion made in lines 255-256 where you state, "4) the NEUT-RI was not affected by renal failure." The authors need to delineate the basis of this claim more explicitly. In line with my previous comment No.2, it's essential to understand that a statistically insignificant result does not necessarily translate to "no effect" or "not affected". The results might be due to the limited power of the study, or it may be that the effect is small and not detectable with the sample size used. Therefore, it's crucial to provide the reasoning
A: we thank the reviewer for this observation. We agree with his comment and acknowledge that absence of evidence is not evidence of absence. We downsized several parts of the text and modified the discussion and the limitation.
In particular, the sentence was modified as follows “the NEUT-RI was not significantly different between patients with and without renal failure”.
In the limitation sections, we added: “Even if the NEUT-RI was not significantly different between patients with and without renal failure it cannot be translated into "no effect". As a secondary objective, the results might be due to the limited power of the study, or it may be that the effect is small and not detectable with the sample size used”.